# Usefulness of PET/CT with Carbon-11-Labeled Methionine in the Diagnosis of Tertiary Hyperparathyroidism

**DOI:** 10.3390/nu15010188

**Published:** 2022-12-30

**Authors:** Maciej Kołodziej, Marek Saracyn, Arkadiusz Lubas, Mirosław Dziuk, Adam Daniel Durma, Jerzy Smoszna, Grzegorz Zelichowski, Stanisław Niemczyk, Grzegorz Kamiński

**Affiliations:** 1Department of Endocrinology and Isotope Therapy, Military Institute of Medicine-National Research Institute, 04-141 Warsaw, Poland; 2Department of Internal Medicine, Nephrology and Dialysis, Military Institute of Medicine-National Research Institute, 04-141 Warsaw, Poland; 3Department of Nuclear Medicine, Military Institute of Medicine-National Research Institute, 04-141 Warsaw, Poland; 4DaVita Dialysis Unit, Wolski Hospital, 01-211 Warsaw, Poland

**Keywords:** PET/CT, tertiary hyperparathyroidism, carbon-11, methionine, hyperparathyroidism

## Abstract

**Introduction**: Tertiary hyperparathyroidism (tHP) may develop in patients treated with hemodialysis or peritoneal dialysis. Parathyroidectomy may result in a significant reduction in the severity of symptoms. For the effective surgical treatment of hyperparathyroidism, proper localization of the parathyroid glands prior to surgery is essential. The sensitivity of scintigraphy in the diagnosis of tHP is lower than in the diagnosis of primary hyperparathyroidism. In recent years, positron emission tomography (PET/CT) has been gaining importance, usually as a complementary technique. **Aim**: The aim of this study was to determine the usefulness of PET/CT with [^11^C]MET in the preoperative localization diagnosis of patients with tertiary hyperparathyroidism caused by chronic kidney disease, in whom first-line diagnostic methods did not allow the localization of pathologically parathyroid glands. **Material and methods**: The study was conducted in a group of 19 adult patients with severe tHP who were resistant or intolerant to non-invasive treatment, with negative results of scintigraphy and ultrasonography of the neck. The study protocol included measurement of the concentration of calcium, phosphorus, and PTH in the blood serum and performing PET/CT with [^11^C]MET. **Results**: A positive result of PET/CT was obtained in 89.5% of the patients (17/19). Parathyroidectomy was performed in 52.9% of the patients (9/17) with positive results of PET/CT with [^11^C]MET, which were fully consistent with the results of the histopathological examinations of the removed parathyroid glands. On this basis, the sensitivity of PET/CT with [^11^C]MET in the preoperative localization diagnosis of patients with tHP was found to be 100%. Multiple lesions were visualized in 57.9% of the patients (11/19). Ectopic lesions were visualized in 21.1% of the patients (4/19). **Conclusions**: PET/CT with [^11^C]MET is a sensitive technique for the second-line preoperative imaging of parathyroid glands in patients with tertiary hyperparathyroidism in whom first-line examinations, such as ultrasound and scintigraphy, has failed.

## 1. Introduction

Primary hyperparathyroidism (pHPT), the condition in which the parathyroid glands produce more parathyroid hormone (PTH) than the body needs, is one of the most common endocrinopathies, and the only method that enables complete recovery is surgery [1,2,3]. Scintigraphy combined with ultrasound is a well-established method in the preoperative localization diagnostics of pHPT [4].

Secondary (sHPT) and tertiary hyperparathyroidism (tHPT) remain a slightly different problem. It is estimated that sHPT occurs in most patients with chronic kidney disease (CKD) when the glomerular filtration rate (GFR) is reduced to approximately 60 mL/min (G3a criterion according to KDIGO) and in almost all patients in higher stages, and tHPT is a consequence of inappropriate sHPT treatment [5]. Worldwide, 1.2 million people died from CKD in 2017, and the global death rate from CKD between 1990 and 2017 increased by 41.5% [6]. The global incidence of CKD in all age groups has increased by 29.3% since 1990 and is estimated to be approximately 9.1, making it a civilization disease. Although precise epidemiological data are not available, it can be assumed that as the number of patients with CKD increases, the incidence of secondary and tertiary hyperparathyroidism also increases.

Despite the introduction of calcimimetics, which allows for partial control of the disease, surgical methods are still the only ones that help fully cure the disease. Scintigraphy and ultrasonography of the parathyroid glands, recognized as first-line diagnostic methods, have limited sensitivity and specificity in the preoperative localization diagnosis of tHPT, mainly due to the frequent presence of multiple and ectopic lesions [7,8,9,10]. Therefore, new methods and radiotracers that allow for a more precise evaluation are being sought. In recent years, there have been an increasing number of reports on the potential role of PET/CT in the diagnosis of both primary and tertiary hyperparathyroidism. One of the radiotracers used in PET/CT can be 18-fluoro-choline ([^18^F]FCH), which is characterized by higher sensitivity and specificity in the diagnosis of primary hyperparathyroidism compared to the first-line imaging methods used so far. Choline is a marker of cell proliferation whose usefulness in the diagnosis of hyperparathyroidism was discovered accidentally in studies of patients diagnosed with prostate cancer. The mechanism of the accumulation of this radiotracer in parathyroid cells is not clear. One of the hypotheses is the increased proliferation of cells in an adenoma and the use of choline, a precursor of phosphatidylcholine, to build cell membranes. Another mechanism that may be responsible for the accumulation of [^18^F]FCH in the parathyroid glands is the stimulation of choline kinase associated with the secretion of PTH [11,12,13]. However, it is not a method with ideal sensitivity and specificity. Another radiotracer used in PET/CT can be 11-carbon-methionine ([^11^C]MET). The mechanism responsible for the increased uptake of [^11^C]MET in hyperactive parathyroid glands is probably related to the use of methionine (as an exogenous amino acid) as one of the substrates for the production of preProPTH (a precursor of PTH). The hormone preProPTH contains 7 methionine residues; therefore, increased production of PTH is associated with an increased demand for methionine and increased accumulation of the methionine-based radiotracer (i.e., [^11^C]MET) in hypertrophied parathyroid glands [14,15]. Its significant role in the diagnosis of pHPT and its higher sensitivity and specificity compared to scintigraphy and ultrasonography have been shown in the literature to date [7,16,17,18]. However, reports on the usefulness of PET/CT with [^11^C]MET in the diagnosis of tHPT are still scarce. For this reason, it seems valuable to establish the role that PET/CT with [^11^C]MET may play in the diagnosis of tHPT.

### Aim

The aim of the study was to determine the usefulness of PET/CT with [^11^C]MET in the preoperative diagnosis of patients with tertiary hyperparathyroidism in whom first-line diagnostic methods did not allow the localization of pathologically changed glands.

## 2. Materials and Methods

The study was approved by the Bioethics Committee at the Military Medical Chamber in Warsaw, Poland (consent No. 161/18). The research was financed under the statutory project of the Military Institute of Medicine-National Research Institute in Warsaw, Poland (grant No. 544/2018). The study group consisted of 19 adult patients with tHPT diagnosed at the Department of Endocrinology and Isotope Therapy of the Military Institute of Medicine-National Research Institute from January 2019 to June 2020. There were 9 women and 10 men in the study group. Of the examined patients, 18 were treated with hemodialysis, and in 1 patient, tHPT persisted after kidney transplantation. In the study group, 3 patients were previously treated surgically for tHPT. The mean age of the patients was 49 ± 16 years. The mean value of body mass index (BMI) was 27.6 ± 6.4 kg/m^2^. 

The inclusion criteria for the study were as follows:
(1)age over 18;(2)severe refractory to conservative treatment (or intolerance to conservative treatment), potentially qualified for parathyroidectomy tHPT (serum PTH concentration above 1000 pg/mL and/or the presence of complications such as persistent itching of the skin, bone pain, calcification of extraosseous tissues), or persistent tertiary hyperparathyroidism after kidney transplantation;(3)negative result of the scintigraphic examination (according to the protocol described below) and negative result of the ultrasound examination of the neck;(4)reading the Clinical Trial Information Sheet and signing Informed Consent to Participate in the Trial;

The exclusion criteria were as follows:
(1)age under 18;(2)no indications for surgical treatment of tertiary hyperparathyroidism;(3)lack of consent of the patient to participate in the study;(4)pregnancy and breastfeeding.

The study protocol included the following:(1)determination of serum calcium and phosphorus levels and serum PTH level in order to confirm tertiary hyperparathyroidism, previously diagnosed in nephrological centers;(2)PET/CT examination with [^11^C]MET according to the protocol described below;(3)comparison of the PET/CT with [^11^C]MET results with postoperative histopathological results of the removed parathyroid glands.

Biochemical determinations were made at the Department of Laboratory Diagnostics of the Military Institute of Medicine-National Research Institute. Total calcium (normal range 2.15–2.55 mmol/L) and serum phosphate (normal range 0.81–1.45 mmol/L) concentrations were determined using the photometric method, and serum PTH (normal range 15–65 pg/mL) concentrations were measured using the electrochemiluminescence (ECLIA) method using Roche Diagnostics reagents.

Scintigraphic examinations were performed on the NM/CT 870DR apparatus by the General Electric Medical System according to the standard protocol used in the Department of Nuclear Medicine of the Military Institute of Medicine-National Research Institute.

Two radiotracers were used for the scintigraphic examination: 110 MBq of [^99m^Tc]NaTcO_4_ and 740 MBq of [^99m^Tc]Tc-MIBI. Planar imaging (subtraction) and two SPECT/CT scans were performed (with an additional delayed SPECT/CT acquisition 150 min after the first SPECT/CT).

Ultrasound examinations were performed at the Department of Endocrinology and Isotope Therapy of the Military Institute of Medicine-National Research Institute on the Siemens Acuson X150 apparatus using a linear transducer with a frequency of 8 MHz.

PET/CT examinations were performed at the Mazovian PET/CT Center *Affidea* using a 64-row Discovery 710 scanner from General Electric Medical Systems. PET/CT acquisition ranged from the top of the eye sockets to the diaphragm; it started 10 min after intravenous administration of [^11^C]MET with an activity of 5 MBq/kg body weight and lasted 9 min, with a time of 3 min for a single bed position (3 bed positions of 3 min each). In the absence of visualization of foci of increased [^11^C]MET accumulation, which potentially might correspond to parathyroid glands, a delayed acquisition was performed 40 min after administration of the radiotracer. The PET/CT images were evaluated by two experienced nuclear medicine specialists using the multimodal Volume Share 5-Advantage Workstation 4.6 from General Electric Medical Systems.

The results obtained were presented as mean with standard deviations and median with extreme values. Nominal variables were presented in the form of numbers with frequency of occurrence. The compliance of the distribution of variables with the normal distribution was checked using the Shapiro–Wilk test. The results of the tests performed were considered significant for the two-sided *p* < 0.05. All statistical analyses were performed using the Statistica v.12 package (Statsoft, Krakow, Poland).

## 3. Results

PET/CT with [^11^C]MET was performed on 19 patients, with negative results for both first-line imaging examinations. In this group, a positive result was obtained in 89.5% of the patients (17/19) (Figure 1 and Figure 2). In 2 patients (10.5%), the result was assessed as inconclusive (both of these patients showed accumulation of the radiotracer in a single lesion, the location and structure of which indicated that it might be a mediastinal lymph node; however, these lesions were not histologically verified due to revealed contraindications for surgery). Multiple lesions were observed in 57.9% of the patients (11/19) (Figure 3). Ectopic lesions were observed in 21.1% of the patients (4/19) (Figure 4). In total, 34 foci of increased accumulation of the radiotracer, considered as parathyroid glands, were observed. Additionally, in 15.8% of the patients (3/19), foci of increased accumulation of the radiotracer in the bones corresponding to brown tumors were observed (Figure 5a,b). Detailed results of the PET/CT examinations with [^11^C]MET are shown in Table 1.

Of the patients with positive PET/CT results with [^11^C]MET, parathyroidectomy was performed in 52.9% (9/17), and the results of histopathological examinations were verified with clinical data. In all patients who underwent parathyroidectomy, there was a concordance between a positive PET/CT result with [^11^C]MET and the histopathological examination in which hypertrophied parathyroid glands were found. Among the remaining 8 (47.1%) patients with a positive PET/CT with [^11^C]MET result, 1 patient did not consent to surgical treatment, 2 were disqualified from surgery by a nephrologist, while 5 patients died while waiting for surgery. The cause of death in 3 patients was coronavirus disease-19 (COVID-19), caused by severe acute respiratory syndrome coronavirus 2 (SARS-CoV-2), while 2 patients died of cardiovascular causes.

The histopathological results obtained after parathyroidectomy revealed the presence of hypertrophied parathyroid glands in all cases, which confirmed clinical, laboratory, and imaging (positive PET/CT results) diagnosis of tertiary hyperparathyroidism. Moreover, clinical improvement and a statistically significant reduction in PTH concentrations were noted in all patients operated upon (on average, from 1538.4 pg/mL to 87 pg/mL, *p* < 0.005). The results of the follow-up examinations are shown in Table 2. In 1 patient, neurosurgical removal of the tumor growing into the spinal canal (assessed as a brown tumor in PET/CT with [^11^C]MET) was also performed. The histopathological examination performed after the neurosurgical treatment confirmed that this lesion was indeed a brown tumor. Two patients, after successful parathyroidectomy, underwent kidney transplantation.

On this basis (pathologically confirmed cases), the sensitivity of PET/CT with [^11^C]MET in the preoperative localization diagnosis of patients with tHPT was found to be 100%. Due to the lack of true-negative and false-positive results in the study group, it was not possible to assess the specificity. This was due to the protocol adopted and approved by the Bioethics Committee, according to which only patients with negative first-line test results qualify for PET/CT.

## 4. Discussion

In the present group of patients with tertiary hyperparathyroidism in whom previous first-line examinations gave a negative result, nearly 89.5% showed a positive result of PET/CT with [^11^C]MET. Moreover, histopathological examinations confirmed the diagnosis of the disease in all patients who underwent parathyroidectomy, allowing for the determination of 100% sensitivity of PET/CT with [^11^C]MET in this group of patients. It should be noted, however, that in the whole group, there were also two results of PET/CT with [^11^C]MET that were assessed as inconclusive. The reason for this assessment was the detection, on PET/CT images, of a faint radiotracer uptake in focal lesions with a location and structure typical of mediastinal lymph nodes (SUV_max_ 1.1 and SUV_max_ 1.3). Histopathological verification in these cases was not possible due to the contraindications for surgery. Because the radiotracer uptake in these lesions was found to be significantly below the radiotracer uptake in the parathyroid glands observed in the other patients (mean SUV_max_ 6.5), it should be assumed that these results were false negative. Assuming that the remaining histopathologically unverified positive results of PET/CT with [^11^C]MET were true positive, the sensitivity of PET/CT with [^11^C]MET in the present group of patients was 89.5%. At the same time, the lack of true-negative and false-positive PET/CT results with [^11^C]MET in the group studied made it impossible to determine specificity.

In a similar group of patients, Rubello et al. obtained positive PET/CT results with [^11^C]MET in 55% (10 out of 18) of patients [19]. In this group, just like in ours, there were dialysis patients with a negative scintigraphic examination result, and the PET/CT examination protocol was comparable to the one used by us. Some of the patients in this group had a positive ultrasound test result, and the mean PTH concentration in this group was 705 pg/mL, which was clearly lower than in our group of patients (mean PTH concentration in our group was 1543.68 pg/mL). It is possible that the Rubello et al. group could include patients with secondary hyperthyroidism or incomplete autonomy of hyperparathyroidism, which may be indicated by a much lower concentration of PTH in their group. This could have led to a much lower percentage of positive PET/CT with [^11^C]MET results. All patients with a positive PET/CT result from this group underwent parathyroidectomy, with positive histopathological verification of all foci of increased accumulation of [^11^C]MET radiotracer revealed in PET/CT. As in our group, the authors found no false-positive PET/CT results. On the other hand, in a Uruguayan study of patients with CKD and tHPT, 88% (15 out of 17) had positive PET/CT with [^11^C]MET results [20]. However, the results of this study by Quagliata et al. were published only in the form of an abstract, so a precise comparative analysis of their study group with ours was not possible. The authors indicated that the mean concentration of PTH in the group was 1096 pg/mL. In our group, all patients were negative for ultrasound and [^99m^Tc]Tc-MIBI scintigraphy, and the activity of the [^11^C]MET radiotracer administered during PET/CT was similar to that used by us. Eight patients from this group (out of 15 with positive PET/CT results) had undergone surgical treatment, and similar to our group, histopathological examination confirmed the presence of hypertrophied parathyroid glands in the location indicated by PET/CT with [^11^C]MET in all operated patients.

Other groups of patients presented in the available literature were not homogeneous. In a study involving 34 patients, Sundin et al. obtained a high percentage (85%) of true-positive, histopathologically verified PET/CT with [^11^C]MET results [21]. However, in this group, only 2 patients had a diagnosis of CKD-related tHPT. Both these patients had a positive PET/CT result. The remaining patients in this group were diagnosed using PET/CT with [^11^C]MET due to MEN1-related (7 patients) or sporadic primary (pHPT) hyperparathyroidism (25 patients). Most of the patients included in the study (25 out of 34 patients) had previously undergone surgery and were diagnosed with recurrent hyperparathyroidism. Due to the lack of false-positive results, the authors assessed the specificity of PET/CT with [^11^C]MET as 100%. An important caveat is the fact that this study used a higher activity of the radiotracer (average 750 ± 141 MBq) and a continuous 45 min acquisition.

In 2004, Otto et al. published a study aimed at determining the optimal protocol for preoperative parathyroid imaging [15]. In the group of 30 patients with hyperparathyroidism, there were 12 with tertiary one. The percentage of positive PET/CT with [^11^C]MET results in these patients was 91.7% (11/12 patients). In this subgroup PET/CT revealed 25 lesions corresponding to the parathyroid glands, which were then positively histopathologically verified as hypertrophied parathyroid glands. Moreover, in this subgroup, no false-positive PET/CT with [^11^C]MET results were found, which is consistent with our study. However, unlike the group presented by us, where all patients who qualified for PET/CT had negative results of the scintigraphy test with [^99m^Tc]Tc-MIBI, in the group of Otto et al., only 2 (out of 12 patients with tHPT) had negative scintigraphic results. The authors also used radiotracer activities much higher than ours (900–1100 MBq). This study also included 18 patients with pHPT (16 with parathyroid adenoma and 2 with parathyroid carcinoma). Among this subgroup in 16 patients (14 with PNP and 2 with cancer), PET/CT with [^11^C]MET was positive. In 13 patients with pHPT, histopathological examination confirmed the presence of an adenoma in the location indicated using PET/CT; in 1 patient with pHPT, the focus of accumulation of the radiotracer visible in PET/CT in the histopathological examination was verified as thyroid adenoma (this lesion also showed increased accumulation of the radiotracer [^99m^Tc]Tc-MIBI in scintigraphy). In both analyzed patients with parathyroid carcinoma, the authors confirmed the compliance of the PET/CT result with [^11^C]MET with the histopathological assessment. In total, in the entire group presented by the authors, 12 patients had negative scintigraphic examination results (10 with pHPT, 2 with tHPT). In 11 of them (91.7%), including 2 patients with tHPT, PET/CT with [^11^C]MET allowed for localization of the parathyroid gland. This percentage is similar to the results of our study, but our group, unlike the group of Otto et al., was homogeneous—it included only tHPT patients with negative ultrasound and scintigraphy results. Therefore, compared with the literature, ours is probably the most numerous homogeneous group of patients with tHPT in whom PET/CT with [^11^C]MET was successfully used after negative first-line imaging results.

There are also reports in the literature about a high percentage of positive PET/CT with [^11^C]MET results in homogeneous groups of patients with pHPT and negative first-line imaging results. Noltes et al. carried out a retrospective analysis of 28 patients with pHPT [17]. In this group, 19 patients had a negative and 9 patients had an uncertain result of [^99m^Tc]Tc-MIBI scintigraphy. At the same time, 7 patients had a negative ultrasound and 6 patients had an uncertain ultrasound result (15 patients had no ultrasound examination). The authors obtained 21 positive PET/CT with [^11^C]MET results (75%), but 3 of them turned out to be false positives because the parathyroid gland previously observed in PET/CT with [^11^C]MET was not located during surgery. However, in all 7 patients with negative PET/CT results, a parathyroid adenoma was identified during surgery. The relatively high percentage of negative PET/CT results obtained by the authors was probably related to the retrospective analysis of examinations performed on two different devices (one hybrid device and one older PET device, allowing only planar imaging) according to two significantly different protocols over a long period of time (2006–2014). The authors estimated the sensitivity of PET/CT with [^11^C]MET in the diagnosis of pHPT to be 72%. Mallikarjuna et al. achieved similar results when assessing the usefulness of various imaging techniques in preoperative localization diagnostics of pHPT [22]. They retrospectively analyzed the results of imaging studies performed on 54 patients treated surgically for pHPT. In 8 of them, PET/CT imaging with [^11^C]MET was performed due to the negative first-line examination results; in 6 patients (75%), it gave a positive result (in 1 patient it was a false-positive result). The sensitivity of PET/CT with [^11^C]MET in this small group of first-line negative patients was estimated to be 71.4%. However, the authors did not describe the exact protocol for PET/CT with [^11^C]MET.

Much worse results were reported by Traub-Weidinger et al. for the group of 15 patients with pHPT, with negative results from the scintigraphic examination and at least one previous surgical procedure in the neck area [23]. In this group, 4 patients had undergone one surgery in the neck (2 patients due to pHPT, and 2 patients due to parathyroid carcinoma), and 11 patients had undergone more than one surgery in the neck (6 patients due to thyroid cancer coexisting with pHPT, and 5 patients due to nodular goiter of the thyroid gland coexisting with pHPT). In the group of 15 PET/CT examinations with [^11^C]MET, a positive result was obtained in only 6 patients (40%), despite the use of a radiotracer activity higher than in our group (700–900 MBq), the use of a contrast enhancement in the tomographic part of the PET/CT, and longer acquisition time (4 min/bed position). The likely reason for such a low percentage of positive results in this group was the previously performed surgeries, which significantly hindered the assessment of PET/CT images. Based on the results of PET/CT with [^11^C]MET, 5 patients from this group underwent another surgical treatment (1 patient did not consent to another surgery). In all of them, it was confirmed that the foci of high accumulation of the radiotracer, observed in PET/CT with [^11^C]MET, were related to pHPT (adenoma was diagnosed in 4 patients, and metastasis of the previously treated parathyroid carcinoma in 1 patient). The authors found no false-positive PET/CT with [^11^C]MET results in this group of patients. The lack of false-positive PET/CT with [^11^C]MET results is consistent with our observations, despite the differences between the two groups. In our study, among patients who underwent PET/CT with [^11^C]MET, three patients had undergone prior surgical treatment, of which 2 were positive for PET/CT.

Reports from recent years show that similarly satisfactory results in the diagnosis of pHPT can be obtained with the [^18^F]FCH radiotracer. In 2021, Bioletto et al. meta-analyzed 22 studies (8 evaluating the diagnostic value of [^11^C]MET and 14 evaluating [^18^F]FCH), stating that in the preoperative diagnosis of pHPT, the sensitivity of [^18^F]FCH is higher than the sensitivity of [^11^C]MET (92% and 80%, *p* < 0.01); however, both radiotracers have a similar positive predictive value (94% for [^18^F]FCH and 95% for [^11^C]MET, *p* = 0.99) [24]. In 2021, Lee et al. conducted a meta-analysis in which they enrolled 8495 patients from 119 studies comparing the use of at least two preoperative imaging techniques in pHPT [25]. By analyzing the area under the cumulative ranking curve (SUCRA) for PET/CT using [^18^F]FCH, PET/CT with [^11^C]MET, planar scintigraphy, single-photon emission computed tomography (SPECT) scintigraphy, as well as ultrasound and 4D-CT, we estimated the probability with which a given imaging technique may be useful for this type of diagnostics. Among the assessed, PET/CT with [^18^F]FCH (SUCRA = 0.9897) obtained the highest value. In order of decreasing SUCRA value, the investigators assessed the usefulness of PET/CT with [^11^C]MET (0.7046), SPECT with [^99m^Tc]Tc-MIBI (0.5465), subtractive planar scintigraphy (0.3241), ultrasound (0.1286), and two-phase planar scintigraphy with [^99m^Tc]Tc-MIBI (0.0585). Remarkably, very good results were obtained for the 4D-CT study (0.7780).

Few reports in the literature indicate the high usefulness of [^18^F]FCH in preoperative diagnostics in patients with tHPT. In a Chinese study from 2019, which included a group of 17 dialysis patients with hyperparathyroidism, Xue et al. assessed the sensitivity of PET/CT with [^18^F]FCH to be 84.1% (with 100% specificity and 86.5% accuracy) [26]. Positive PET/CT results with [^18^F]FCH were obtained in all patients included in the study, and no false positives were found. All patients in this group (unlike the patients in our group) also had a positive result from the scintigraphic examination with [^99m^Tc]Tc-MIBI, and the mean concentration of PTH in the whole group was 2191.93 ± 840.85 pg/mL, which is a higher value than the average concentration of PTH in our group.

The limitation of the present study is the relatively small size of the group, although, as shown earlier, it is probably the largest group of tHPT patients with negative ultrasound and scintigraphy results undergoing PET/CT imaging with [^11^C]MET. Collecting a large, homogeneous group of patients with tHPT was not an easy task, despite the fact that the recruitment lasted 2 years, and the study included patients from many dialysis centers in Masovian Voivodship (a region with a population of almost 5.5 million people). Difficulties in research on tHPT also result from objective reasons, such as a relatively small population size of patients with end-stage renal disease compared to other diseases. Another limitation of the study was the lack of histopathological verification in all participating patients. This was related either to the refusal to consent to surgery or to contraindications to such treatment. The special time of the global COVID-19 pandemic, during which this work took place and patients died of SARS-CoV-2 infections, also played a significant role, especially in such a sensitive population as patients with end-stage renal disease.

The greatest advantage of the present study is the high percentage of positive PET/CT with [^11^C]MET results in the group of patients with negative first-line imaging results. The group of these patients, in the light of the standards of management of hyperparathyroidism currently adopted in Poland and financing from public funds, is deprived of access to more advanced diagnostic techniques. The results obtained may have a direct, positive impact on clinical and therapeutic management and may also contribute to their modification in the future. A great advantage of this study is the fact that it was conducted on a homogeneous group of patients with tHPT, which is probably the largest in the available literature. The available literature is lacking the most from studies assessing the possibilities of new imaging diagnostic methods in this group (tHPT patients). The value of first-line diagnostic methods in this group of patients is limited, and the availability of the latest methods is still difficult.

Due to the short half-life (T_1/2_ = 20.4 min), [^11^C]MET is a radiotracer used mainly in centers equipped with cyclotrons. According to EANM Guidelines, PET/CT with [^11^C]MET is one of the second-line diagnostic methods, next to PET/CT with [^18^F]FCH [4]. Undeniably, PET/CT with [^18^F]FCH is technically easier to perform, while PET/CT with [^11^C]MET, compared to [^18^F]FCH, is characterized by lower patient exposure to ionizing radiation [4]. It is estimated that the dose absorbed per radiotracer activity unit in PET/CT examination with [^11^C]MET is 0.00549 mSv/MBq, while in PET/CT examination with [^18^F]FCH, it is 0.02 mSv/MBq (almost four times higher).

## 5. Conclusions

The results obtained allow us to conclude that PET/CT with [^11^C]MET is a sensitive technique for the second-line preoperative imaging of the parathyroid glands in patients with tertiary hyperparathyroidism in whom it was not possible to locate them in first-line examinations, such as ultrasonography and scintigraphy.

## Figures and Tables

**Figure 1 nutrients-15-00188-f001:**
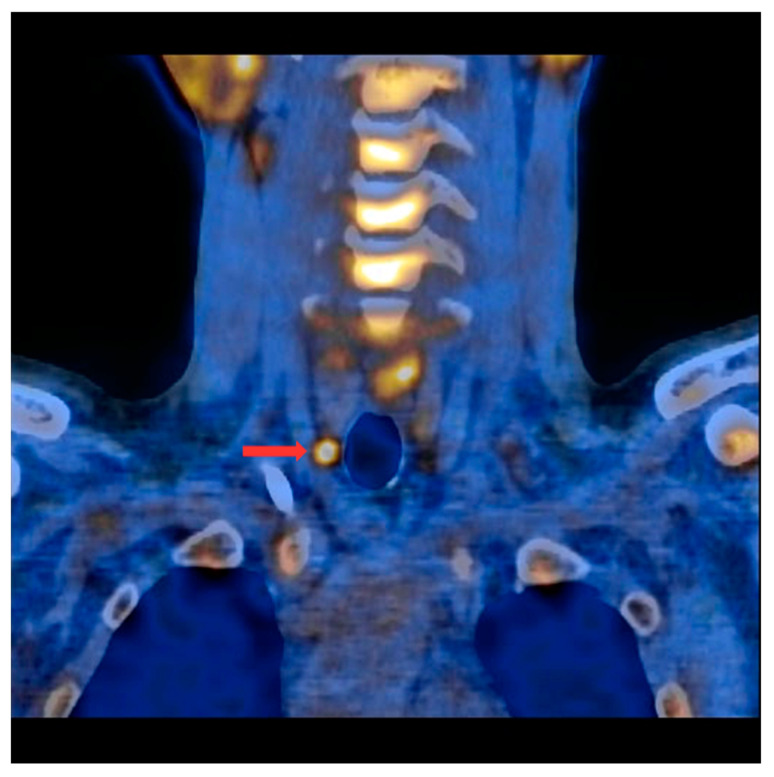
PET/CT with [^11^C]MET—frontal projection. Visible focus of increased radiotracer accumulation (red arrow) corresponding to the parathyroid gland below the right thyroid lobe.

**Figure 2 nutrients-15-00188-f002:**
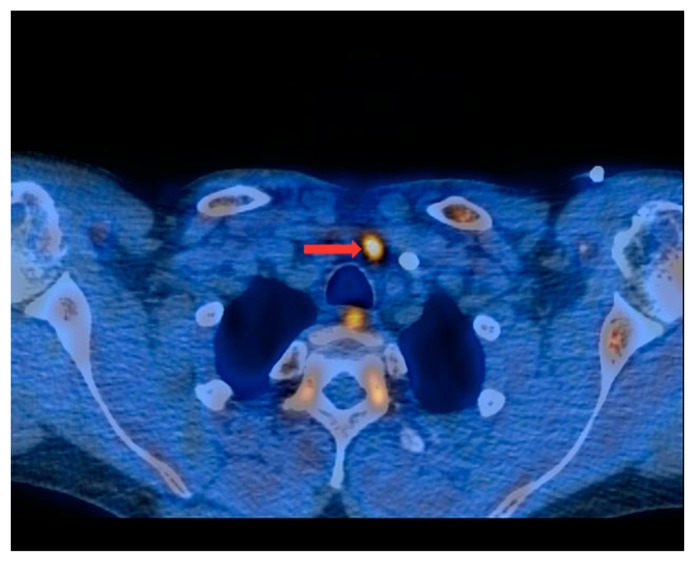
PET/CT with [^11^C]MET—axial projection. Visible focus of increased radiotracer accumulation (red arrow) corresponding to the parathyroid gland in front of the trachea, at the left-hand side of the midline of the body.

**Figure 3 nutrients-15-00188-f003:**
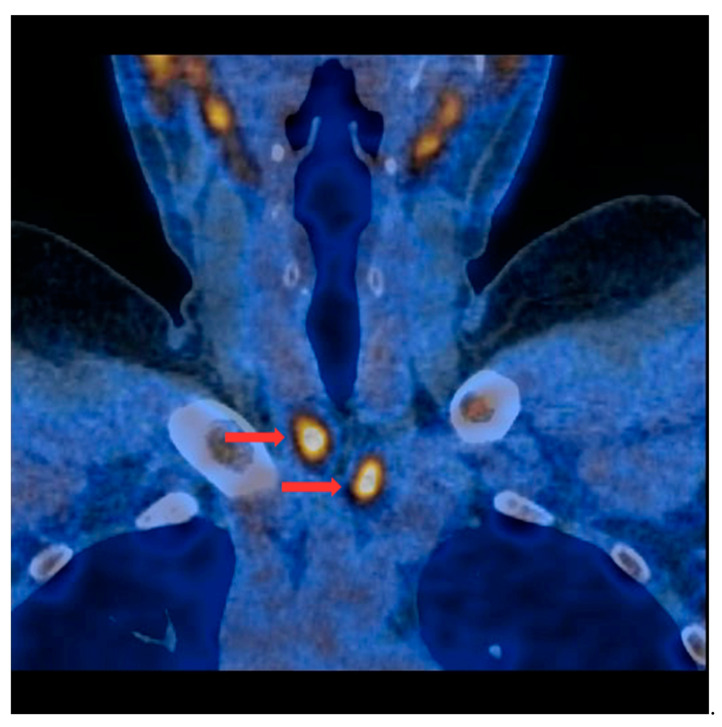
PET/CT with [^11^C]MET—frontal projection. Two visible foci of increased radiotracer accumulation (red arrows) corresponding to the parathyroid glands (multiple lesions) below both thyroid lobes.

**Figure 4 nutrients-15-00188-f004:**
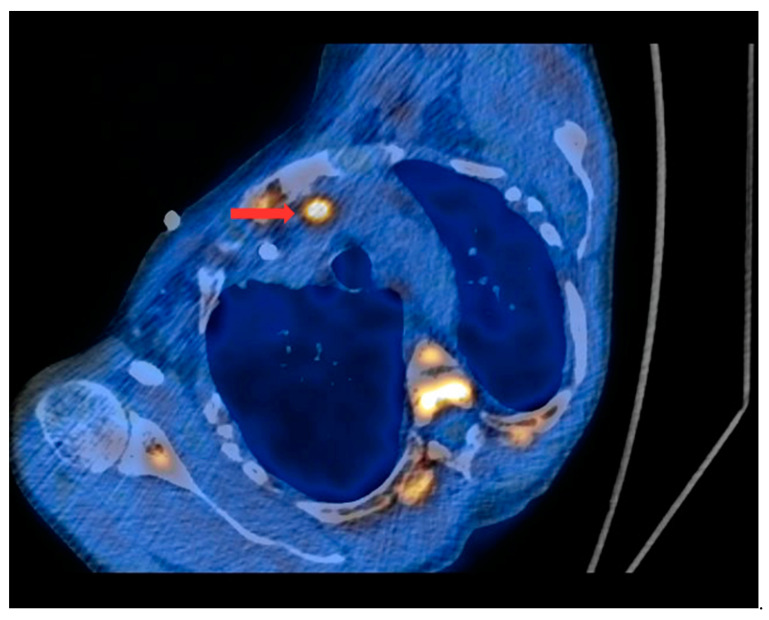
PET/CT with [^11^C]MET—axial projection. During the examination, the patient was laid at her left side due to the forced body position caused by numerous fractures and a brown tumor infiltrating the spinal canal (as a result of tHPT). Visible focus of increased radiotracer uptake (red arrow) in the lesion corresponding to the ectopic parathyroid gland, located in the left superior mediastinum retrosternally, and anteriorly from the initial segment of the aortic arch.

**Figure 5 nutrients-15-00188-f005:**
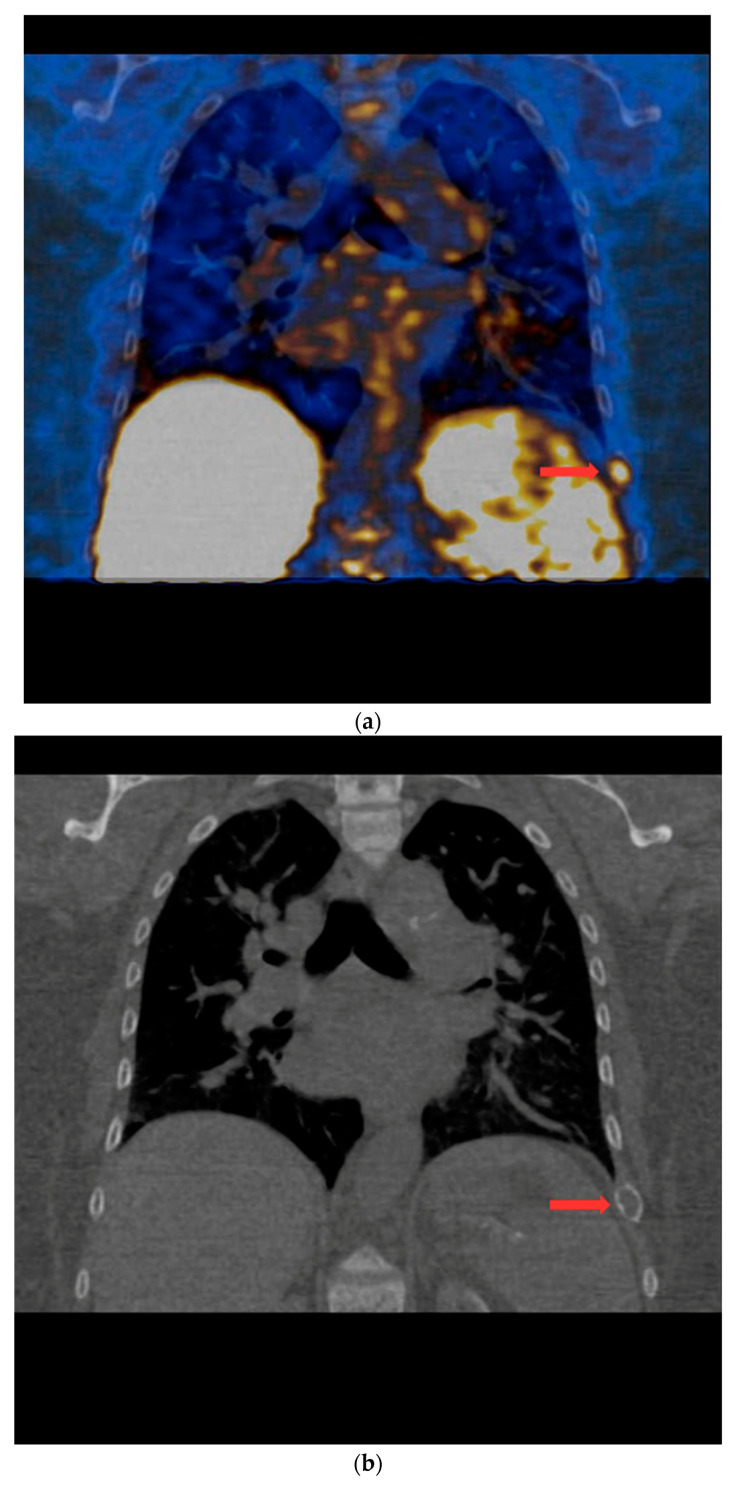
(**a**) PET/CT with [^11^C] MET—frontal projection. Visible focal increased radiotracer uptake (red arrow) corresponding to the brown tumor in the middle part of the left 9th rib. (**b**) Low-dose CT scan performed during PET/CT with [^11^C]MET in the same patient. In the location of the focus of increased accumulation of the radiotracer corresponding to the brown tumor of the left 9th rib, the distention of the contour of this rib is visible (red arrow).

**Table 1 nutrients-15-00188-t001:** PET/CT with [^11^C]MET results.

Patient	Sex	Age (year)	PET/CT Result	Number of Lesions	Ectopic Lesions	Brown Tumors
01	M	39	+	2	no	no
02	F	24	+	2	no	no
03	F	42	+	4	yes	yes
04	F	42	+	2	no	no
05	M	53	+	1	no	no
06	M	26	+	3	no	no
07	F	85	-	0	no	no
08	M	52	+	1	no	no
09	M	59	+	2	no	no
10	M	64	+	1	no	yes
11	M	31	+	1	no	no
12	F	71	+	3	no	no
13	F	66	+	2	no	no
14	F	48	+	1	no	yes
15	M	50	-	0	no	no
16	M	32	+	3	yes	no
17	F	50	+	1	no	no
18	M	52	+	3	yes	no
19	F	62	+	2	yes	no

The sign “-” means an inconclusive result of PET/CT; the sign “+” means a positive result of PET/CT; M—male, F—female.

**Table 2 nutrients-15-00188-t002:** Serum PTH concentration before PET/CT with [^11^C]MET and after surgical treatment.

Patient	PTH before Surgery [pg/mL]	PTH after Surgery [pg/mL]
01 ^(A)^	1123	12.9
02 ^(A)^	2700	11
03	1638	107.7
04	1738	7.7
06 ^(B)^	466	7.5
11	1352	12.8
13	1446	<1.2
14	1180	297.3
16	2202	238.8
mean ± SD	1538.4 ± 645.7	87 ± 117.8 *
median	1446	12.9

^(A)^ patient who qualified for kidney transplantation after parathyroidectomy; ^(B)^ patient with tHPT persisting after kidney transplantation; PTH—parathyroid hormone; SD—standard deviation; (*) distribution does not meet the conditions of normal distribution.

## Data Availability

Data sharing is not applicable to this article.

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
