# Peer review of "Usefulness of PET/CT with Carbon-11-Labeled Methionine in the Diagnosis of Tertiary Hyperparathyroidism"

_nutrients, 2022, doi:10.3390/nu15010188_

Round 1

Reviewer 1 Report

This article is a very significant research. The results showed PET/CT with [11C]MET is a sensitive technique of the second-line pre-operative imaging of the parathyroid glands in patients with tertiary hyperparathyroidism. However the following questions need to be explained or corrected.

1. Highlight the Introduction, Aim, Material and Methods, Results, and Conclusion in the Abstract.

2. “PET/CT and [11C]MET” should be "PET/CT with [11C]MET" in The study protocol included part of page 3. And please check other spell errors.

3.In this study, two patients with high uptake was not histologically verified due to revealed contraindications for surgery. Can these lesions be divided into ectopic lesions or false positive if they were lymph nodes?

4. What's the mechanics of [11C]MET uptake? Please discuss the mechanics difference between  [11C]MET and [18F]FCH to explain their value.

5. Because of the limited image resolution of PET/CT, is it any possible that some smaller lesions can not be detected? 

Author Response

Pleas see the attachment

Reviewer 2 Report

The authors present a study that examines the role of 11C Methionine PET/CT in patients that have had negative imaging findings on ultrasound and conventional scintigraphy in the localization ot parathyroid adenoma and hyperplasia in tertiary hyperparathyroidism, The authors establish their inclusion criteria and exclusion crtiria and select their patient and describe the method for imaging. The compared the positive foci with histopathology for patient who had surgey

A few comment-

1. The abstract is rather long and would benefit from being more focussed on the study eg The first two sentenced could be removed and left in the introduction and authors coult start with Tertiary hyperthyroidism.... (3rd sentencence). Similarly other aspect can be summarized to provide a succint abstract.

2. Under introduction in the first sentence the word 'inadequate' is no appropriate and 'inappropriate' may better fit that context.

3. Could the author please consider replacing civilization disease. Ther are many patients with CKD in developing countries on dialysis not sure tertiary hyperparathyroidism can be classified as a such based on a statistic from Poland.

4. Under introduction authors must visit why PET?CT with C11 or methionine has a better yeilld in tertiary hyperpathyroidism it term of differences in spatial resolution due to conventional scintigraphy and PET/CT as with SPECT/CT conventional scintigraphy also detects ectopic and multiple glands. The reference the used to suppot the limited sensitivity and specificity (reference 7) comes from an article from 1996 wheren the SPECT systems were not as developed as today and very little hybid imaging was done

5. Under methods- 'approval from ethics 'is more conventional than writing 'a positive opinion form ...' please consider changing.

6. Under methods it may be useful to know whether the conventional scintigraphy for which parathyroid hyperplasia could not be identified was a SPECT/CT and whether they employed dual tracer or dual time or both methods.

7. Could the authors explain why the used 1000 as cut off for hyperparathyroidism as in this study and other studies the author found patient with tertiary hyperparathyroidism not that high? Would the author recommend doing imaging for tertiary hyperparathyroidism in that scenaria

8. The manuscript may benefit from a proof reader with native english (moderate) eg, the study patients diagnoses due to - the word with would have been more appropirate here or the word revealed before contraindications under discussion.

9. The use of bed position is more conventional than table position under the acquisition protocol of PET/CT for the sake of reading audience it is better to use the more familiar term please consider changing.

10. The placement of the words figure 1 and Figure 2 may be confusing to the reading audience please place at the end of the 2nd sentence under results not in the middle of the sentence.

11. For table 2- under lengend authors may want to explain what complication this patient had apart from persistent hyperparathyroidism after renal transplant seeing the PTH was not more than a 1000 pg/ml

12. Since the localization of parathyroid lesions was done for surgical control will be interesting to know why 8 patients had no histopathological conffirmation (19- (2 negative scans +9 with histology in table 2) = 8 ie 42% of patients.

13. In terms of limitation of study authors must please discuss how the short half-life of 20 minuted would limit the use of 11C methionine to site with an onsite cyclotron which may hinder its widespread applicability hence the us of 18F based PET tracer more likely to have widespread applicability.

14. Under the discussion 2 references came from much earliar 1996 and 2004 reference 14 and 15 and protocols have changed and equioement improved since then authors must consider mentioning this or removing from discussion.

Round 2

Reviewer 2 Report

The authors have responded to the querries and the manuscript